# Penetrating the Fog: the Path to Efficient CNN Models

## Abstract

With the increasing demand to deploy convolutional neural networks (CNNs) on mobile platforms, the sparse kernel approach was proposed, which could save more parameters than the standard convolution while maintaining accuracy. However, despite the great potential, no prior research has pointed out how to craft an sparse kernel design with such potential (i.e., effective design), and all prior works just adopt simple combinations of existing sparse kernels such as group convolution. Meanwhile due to the large design space it is also impossible to try all combinations of existing sparse kernels. In this paper, we are the first in the field to consider how to craft an effective sparse kernel design by eliminating the large design space. Specifically, we present a sparse kernel scheme to illustrate how to reduce the space from three aspects. First, in terms of composition we remove designs composed of repeated layers. Second, to remove designs with large accuracy degradation, we find an unified property named *information field* behind various sparse kernel designs, which could directly indicate the final accuracy. Last, we remove designs in two cases where a better parameter efficiency could be achieved. Additionally, we provide detailed efficiency analysis on the final 4 designs in our scheme. Experimental results validate the idea of our scheme by showing that our scheme is able to find designs which are more efficient in using parameters and computation with similar or higher accuracy.

## 1 Introduction

CNNs have achieved unprecedented success in visual recognition tasks. The development of mobile devices drives the increasing demand to deploy these deep networks on mobile platforms such as cell phones and self-driving cars. However, CNNs are usually resource-intensive, making them difficult to deploy on these memory-constrained and energy-limited platforms.

To enable the deployment, one intuitive idea is to reduce the model size. Model compression is the major research trend for it. Previously several techniques have been proposed, including pruning (LeCun et al., 1990), quantization (Soudry et al., 2014) and low rank approximation (Denton et al., 2014). Though these approaches can can offer a reasonable parameter reduction with minor accuracy degradation, they suffer from the three drawbacks: 1) the irregular network structure after compression, which limits performance and throughput on GPU; 2) the increased training complexity due to the additional compression or re-training process; and 3) the heuristic compression ratios depending on networks, which cannot be precisely controlled.

Recently the sparse kernel approach was proposed to mitigate these problems by directly training networks using structural (large granularity) sparse convolutional kernels with fixed compression ratios. The idea of sparse kernel was originally proposed as different types of convolutional approach. Later researchers explore their usages in the context of CNNs by combining some of these sparse kernels to save parameters/computation against the standard convolution. For example, MobileNets (Howard et al., 2017) realize 7x parameter savings with only 1% accuracy loss by adopting the combination of two sparse kernels, depthwise convolution (Sifre & Mallat, 2014) and pointwise convoluiton (Lin et al., 2013), to replace the standard convolution in their networks.

However, despite the great potential with sparse kernel approach to save parameters/computation while maintaining accuracy, it is still mysterious in the field regarding how to craft an sparse kernel design with such potential (i.e., effective sparse kernel design). Prior works like MobileNet (Howard et al., 2017) and Xception (Chollet, 2016) just adopt simple combinations of existing sparse kernels, and no one really points out the reasons why they choose such kind of design. Meanwhile, it has been a long-existing question in the field whether there is any other sparse kernel design that is more efficient than all state-of-the-art ones while also maintaining a similar accuracy with the standard convolution.

To answer this question, a native idea is to try all possible combinations and get the final accuracy for each of them. Unfortunately, the number of combination will grow exponentially with the number of kernels in a design, and thus it is infeasible to train each of them. Specifically, even if we limit the design space to four common types of sparse kernels – group convolution (Krizhevsky et al., 2012), depthwise convolution (Sifre & Mallat, 2014), pointwise convolution (Lin et al., 2013) and pointwise group convolution (Zhang et al., 2017) – the total number of possible combinations would be $4^k$, given that $k$ is the number of sparse kernels we allow to use in a design (note that each sparse kernel can appear more than once in a design).

In this paper, we craft the effective sparse kernel design by efficiently eliminating poor candidates from the large design space. Specifically, we reduce the design space from three aspects: composition, performance and efficiency. First, observing that in normal CNNs it is quite common to have multiple blocks which contain repeated patterns such as layers or structures, we eliminate the design space by ignoring the combinations including repeated patterns. Second, realizing that removing designs with large accuracy degradation would significantly reduce the design space, we identify a easily measurable quantity named *information field* behind various sparse kernel designs, which is closely related to the model accuracy. We get rid of designs that lead to a smaller *information field* compared to the standard convolution model. Last, in order to achieve a better parameter efficiency, we remove redundant sparse kernels in a design if the same size of *information field* is already retained by other sparse kernels in the design. With all aforementioned knowledge, we present a sparse kernel scheme that incorporates the final four different designs manually reduced from the original design space.

Additionally, in practice, researchers would also like to select the most parameter/computation efficient sparse kernel designs based on their needs, which drives the demand to study the efficiency for different sparse kernel designs. Previously no research has investigated on the efficiency for any sparse kernel design. In this paper, three aspects of efficiency are addressed for each of the sparse kernel designs in our scheme: 1) what are the factors which could affect the efficiency for each design? 2) how does each factor affect the efficiency alone? 3) when is the best efficiency achieved combining all these factors in different real situations?

Besides, we show that the accuracy of models composed of new designs in our scheme are better than that of all state-of-the-art methods under the same constraint of parameters, which implies that more efficient designs are constructed by our scheme and again validates the effectiveness of our idea.

The contributions of our paper can be summarized as follows:

- We are the first in the field to point out that the *information field* is the key for the sparse kernel designs. Meanwhile we observe the model accuracy is positively correlated to the size of the *information field*.
- We present a sparse kernel scheme to illustrate how to eliminate the original design space from three aspects and incorporate the final 4 types of designs along with rigorous mathematical foundation on the efficiency.
- We provide some potential network designs which are in the scope of our scheme and have not been explored yet and show that they could have superior performances.

## 2 Preliminaries

We first give a brief introduction to the standard convolution and the four common styles of sparse kernels.

### 2.1 Standard Convolution

Standard convolution is the basic component in most CNN models, kernels of which can be described as a 4-dimensional tensor: $W \in \mathbb{R}^{C \times X \times Y \times F}$, where $C$ and $F$ are the numbers of the input and the output channels and $X$ and $Y$ are the spatial dimensions of the kernels. Let $I \in \mathbb{R}^{C \times U \times V}$ be the input tensor, where $U$ and $V$ denote the spatial dimensions of the feature maps. Therefore, the output activation at the output feature map $f$ and the spatial location $(x, y)$ can be expressed as,

$$T(f, x, y) = \sum_{c=1}^{C} \sum_{x'=1}^{X} \sum_{y'=1}^{Y} I(c, x - x', y - y') W(c, x', y', f)$$

### 2.2 Group Convolution

Group convolution is first used in AlexNet (Krizhevsky et al., 2012) for distributing the model over two GPUs. The idea of it is to split both input and output channels into disjoint groups and each

output group is connected to a single input group and vice versa. By doing so, each output channel will only depend on a fraction of input channels instead of the entire ones, thus a large amount of parameters and computation could be saved. Considering the number of group as $M$, the output activation $(f, x, y)$ can be calculated as,

$$T(f, x, y) = \sum_{c'=1}^{C/M} \sum_{x'=1}^{X} \sum_{y'=1}^{Y} I(\frac{C}{M}\lfloor \frac{f-1}{\frac{F}{M}} \rfloor + c', x - x', y - y')W(c', x', y', f)$$

### 2.3 DEPTHWISE CONVOLUTION

The idea of depthwise convolution is similar to the group convolution, both of which sparsifies kernels in the channel extent. In fact, depthwise convolution can be regarded as an extreme case of group convolution when the number of groups is exactly the same with the number of input channels. Also notice that in practice usually the number of channels does not change after the depthwise convolution is applied. Thus, the equation above can be further rewritten as,

$$T(f, x, y) = \sum_{x'=1}^{X} \sum_{y'=1}^{Y} I(f, x - x', y - y')W(x', y', f)$$

### 2.4 POINTWISE CONVOLUTION

Pointwise convolution is actually a $1 \times 1$ standard convolution. Different from the group convolution, pointwise convolution achieves the sparsity over the spatial extent by using kernels with $1 \times 1$ spatial size. Similarly, the equation below shows how to calculate one output activation from the pointwise convolution in detail,

$$T(f, x, y) = \sum_{c=1}^{C} I(c, x, y)W(c, f)$$

### 2.5 POINTWISE GROUP CONVOLUTION

To sparsify kernels in both the channel and the spatial extents, the group convolution can be combined together with the pointwise convolution, i.e., pointwise group convolution. Besides the use of $1 \times 1$ spatial kernel size, in pointwise group convolution each output channel will also depend on a portion of input channels. The specific calculations for one output activation can be found from the equation below,

$$T(f, x, y) = \sum_{c'=1}^{C/M} I(\frac{C}{M}\lfloor \frac{f-1}{\frac{F}{M}} \rfloor + c', x, y)W(c', f)$$

## 3 SPARSE KERNEL SCHEME

Recall that the total number of combinations will grow exponentially with the number of kernels in a design, which could result in a large design space. In this paper, we craft the effective sparse kernel design (i.e., design that consumes less parameters but maintains accuracy with the standard convolution) by efficiently examining the design space.

Specifically, first we determine the initial design space by setting the maximum number of sparse kernels (length). To decide this number, two aspects are considered: 1) in order to give the potential to find more efficient designs which have not been explored yet, the maximum length of sparse kernel design should be greater than the numbers of all state-of-the-art ones; 2) it is also obvious that the greater length is more likely to consume more parameters, which contradicts our goal to find more efficient designs. Therefore combining the two aspects together, we set the maximum length to 6, which is not only greater than the largest number (i.e., 3) in all current designs, but also makes designs with the maximum length could still be able to be more efficient than the standard convolution.

### 3.1 Reduce the Design Space

We then start to reduce the design space from three aspects: composition, performance and efficiency. In the following paragraphs, we will introduce the three aspects in detail.

**Composition.** The overall layout in CNNs provides a good insight for us to quickly reduce the design space. Specifically, in normal CNNs it is quite common to have multiple stages/blocks which contain repeated patterns such as layers or structures. For example, in both VGG (Simonyan & Zisserman, 2014) and ResNet (He et al., 2016a) there are 4 stages and inside each stage are several same repeated layers. Inspired by the fact, when we replace the standard convolution using various sparse kernel designs intuitively there is no need to add these repeated patterns to the original place of each standard convolutional layer. For example, suppose there are three types of sparse kernels, A, B and C, then the following combinations should be removed as containing repeated patterns: AAAAAA, ABABAB and ABCABC. AAAAAA contains the repeated pattern of A, while ABABAB and ABCABC have the patterns of AB and ABC respectively.

Repeated patterns are also easy to detect, which makes the entire process extremely fast. To find such patterns, we can use the regular expression matching. The corresponding expression for the matched combinations should be $(.+?)\mathbf{1}+$, where $(.+?)$ denotes the first capturing group which contains at least one character, but as few as possible, and $\mathbf{1}+$ means try to match the same character(s) as most recently matched by the first group as many times as possible. As a result, we can efficiently eliminate the design space with the help of the regular expression.

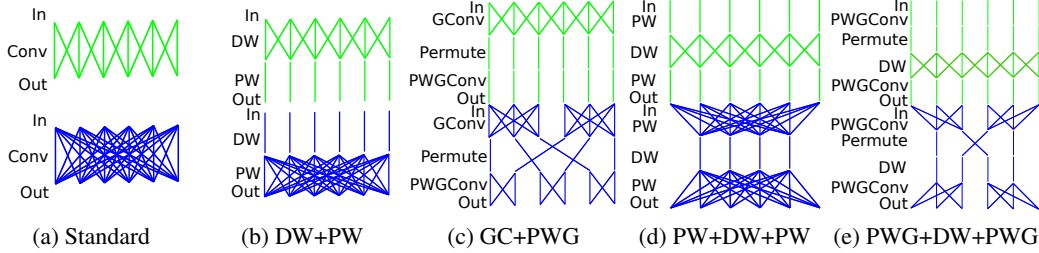

| (a) Standard | (b) DW+PW | (c) GC+PWG | (d) PW+DW+PW | (e) PWG+DW+PWG |

Figure 1: Spatial and channel dependency of the standard convolution and four different sparse kernel designs. The spatial kernel size is $3 \times 3$. Green edges denote the spatial dependency of output activation and blue edges represent the channel dependency.

**Performance.** There are lots of sparse kernel designs that could result in large accuracy degradation, which gives us another opportunity to greatly reduce the design space. To get rid of them, we need an easily measurable (i.e., no training) property behind various designs that could directly indicate the final accuracy. Fortunately, after analyzing many prior works and conducting many experimental studies, we do find such property. We name it *information field*.

**Definition 1.** *(Information Field) Information field is the area in input tensor which one or more convolutional layers use to generate one output activation. For one output tensor, sizes of information fields for all activations are usually the same.*

Figure 1a shows the spatial and channel dependency for the standard convolution, from which we can also find out the size of *information field*. Assuming the spatial kernel size is $3 \times 3$, starting from any output node in the figure we can see that in terms of the channel dimension each output channel will connect to all input channels and for the spatial dimensions one output activation will depend on activations inside a $3 \times 3$ spatial area. Therefore the *information field* for the standard convolution will be (3, 3, C) where C is the number of input channels.

We find that *information field* is the key behind all sparse kernel designs, and also observe the model accuracy is positively correlated to the size of *information field*, the idea of which is also validated by later experiments in Section 4.2.

With the help of *information field*, sparse kernel designs that would result in large accuracy degradation could be easily removed from the original design space without actually training the models. Specifically, first for each design we calculate the size of *information field* by adding up it sequentially from the leftmost kernel to the rightmost one. For example, we use a three-dimensional vector, (1,1,1), to represent the initial values of *information field* on three different dimensions (i.e., two spatial dimensions and one channel dimension), then corresponding values of the vector will be updated based on the known properties of the sparse kernel encountered. After the rightmost kernel, the final

vector we get will be the size of *information field* for the design. Finally we compare it with that of the standard convolution. If the two sizes are the same, we will keep the design, otherwise we will simply discard it. For instance, the design composed of one depthwise convolution will be removed since the *information field* of it only contains one channel area instead of the full channel space from the standard convolution.

**Efficiency.**    Considering a better parameter efficiency more designs could be removed. In paragraphs above, we only eliminate designs in terms of the accuracy via checking the size of *information field*. Recall that our goal is also to find efficient designs. Thus, while ensuring the accuracy we also need to take the efficiency into consideration. In fact, there are two cases that could worsen the efficiency and should be regarded as redundant designs: 1) it can be easily verified that the size of *information field* will never decrease when passing through sparse kernels in a design, thus there could be one situation that after one kernel, the size of *information field* still remains the same, which means the kernel does not help with regards to the *information field* even if the final size is the same as the standard convolution; 2) it is also possible that the same size of *information field* with the standard convolution is already retained by a fraction of sparse kernels in a design, in which case, other kernels can also be considered as not contributing to the *information field*. In terms of parameter efficiency designs in both of the two cases contain non-contributed kernels, therefore we can remove them from the original design space.

To detect designs within the two cases, we introduce a early-stop mechanism during the process to check the size of *information field* above. Specifically, as per the two cases we check two things when adding up *information field* from the leftmost kernel in a design: 1) we record the size of *information field* before entering each kernel and compare it with the new size calculated after that kernel. If the two sizes are the same, we stop adding up *information field* for the design and directly go to the next one; 2) we add another conditional check every time we get a new size of *information field*. If the size is still less than or equal to that of the standard convolution, we will continue to add up *information field* from the next kernel, otherwise we will stop and go to the next design.

With all aforementioned knowledge, we manually reduce the original design space $(4^1 + 4^2 + \cdots + 4^6)$ to 4 different types of sparse kernel designs[1]. In the next section we will present the 4 final designs respectively.

Also notice that other techniques to save parameters such as bottleneck structure (He et al., 2016a) appear to be complimentary to our approach, which can be combined together to further improve parameter efficiency while maintaining accuracy. To validate this idea, we also consider the bottleneck structure when reducing the design space.

## 3.2    FINAL SPARSE KERNEL DESIGNS

**Depthwise Convolution + Pointwise Convolution.**    Unlike the standard convolution which combines spatial and channel information together to calculate the output, the combination of depthwise convolution (DW) and pointwise convolution (PW) split the two kinds of information and deal with them separately. The output activation at location $(f, x, y)$ can be written as

$$T(f,x,y) = \sum_{c=1}^{C} [\sum_{x'=1}^{X} \sum_{y'=1}^{Y} I(c, x-x', y-y')W_1(c, x', y')]W_2(c, f),$$

where $W_1$ and $W_2$ correspond to the kernels of depthwise convolution and pointwise convolution respectively. The dependency of such design is depicted in Figure 1b, from which we can easily verify that the size of *information field* is the same with the standard convolution.

**Group Convolution + Pointwise Group Convolution.**    The combination of group convolution (GC) and pointwise group convolution (PWG) can be regarded as an extension for the design above, where group convolution is applied on the pointwise convolution. However, simply using pointwise group convolution would reduce the size of *information field* on the channel dimension since depthwise convolution will not deal with any channel information. To recover the *information field* depthwise convolution is replaced with the group convolution. Meanwhile channel permutation should be added between the two layers. Assuming the number of channels does not change after the

---

[1]During the process to eliminate the design space, we allow channel permutation within the designs, and when a group convolution is encountered, we will try all possible numbers of groups to calculate the size of *information field*. As long as there is one group number that can pass the entire process, we will keep the design. In case there are multiple group numbers passing the process, we will consider them as same design.

first group convolution, the output activation can be calculated as

$$T(f,x,y) = \sum_{k'=1}^{C/N}[\sum_{c'=1}^{C/M}\sum_{x'=1}^{X}\sum_{y'=1}^{Y}I(\frac{C}{M}\lfloor\frac{k-1}{\frac{C}{M}}\rfloor + c', x-x', y-y')W_1(c',x',y',k)]W_2(k',f),$$

where $k = \frac{C}{N}\lfloor\frac{f-1}{\frac{F}{N}}\rfloor + k'$, $M$ and $N$ denote numbers of groups for group convolution and pointwise group convolution and $W_1$ and $W_2$ correspond to the kernels of group convolution and pointwise group convolution respectively. Figure 1c shows the *information field* of this design clearly.

**Pointwise Convolution + Depthwise Convolution + Pointwise Convolution.** Although two pointwise convolutions do not ensure a better efficiency in our scheme, the combination with bottleneck structure can help ease the problem, which makes it survive as one of the last designs. Following the normal practice we set bottleneck ratio to $1 : 4$, which implies the ratio of bottleneck channels to output channels. Also notice that more parameters could be saved if we place the depthwise convolution between the two pointwise convolutions since now depthwise convolution would only apply on a reduced number of channels. As a result, the output activation $T(f,x,y)$ is calculated as

$$T(f,x,y) = \sum_{k=1}^{K}[\sum_{x'=1}^{X}\sum_{y'=1}^{Y}[\sum_{c=1}^{C}I(c,x-x',y-y')W_1(c,k)]W_2(k,x',y')]W_3(k,f),$$

where $K$ denote the number of bottleneck channels and $W_1$, $W_2$ and $W_3$ correspond to the kernels of first pointwise convolution, depthwise convolution and second pointwise convolution respectively. Along with the equation Figure 1d shows that the *information field* of such design is same with the standard convolution.

**Pointwise Group Convolution + Depthwise Convolution + Pointwise Group Convolution.** The combination of two pointwise group convolutions and one depthwise convolution can also ensure the same size of *information field*. Similarly, channel permutation is needed. The bottleneck structure is also adopted to achieve a better efficiency. The output activation is calculated as

$$T(f,x,y) = \sum_{k'=1}^{K/N}[\sum_{x'=1}^{X}\sum_{y'=1}^{Y}[\sum_{c'=1}^{C/M}I(\frac{C}{M}\lfloor\frac{k-1}{\frac{K}{M}}\rfloor+c', x-x', y-y')W_1(c',k)]W_2(k,x',y')]W_3(k',f),$$

where $k = \frac{K}{N}\lfloor\frac{f-1}{\frac{F}{N}}\rfloor + k'$, $K$, $M$ and $N$ represent the number of bottleneck channels and numbers of groups for first pointwise group convolution and second pointwise group convolution and $W_1$, $W_2$ and $W_3$ correspond to the kernels of first pointwise group convolution, depthwise convolution and second pointwise group convolution respectively. Both the equation and Figure 1e could verify the same size of *information field* with the standard convolution.

### 3.3 EFFICIENCY ANALYSIS

In addition, we find that the efficiency for different designs in our scheme do not always overlap. Thus to save the pain for researchers to find the most parameter/computation efficient designs based on their needs, we study the efficiency for each of the designs. Specifically, we consider two real situations which are frequently encountered by researchers when applying sparse kernel designs (i.e., given the input and the output for a layer and given the total number of parameters for a layer) and give accurate conditions when the best efficiency could be achieved.

#### 3.3.1 DEPTHWISE CONVOLUTION + POINTWISE CONVOLUTION.

**Efficiency given the input and the output.** Given the numbers of input and output channels $C$ and $F$. The total number of parameters after applying this design is $9C + CF$, and the number of parameters for standard convolution is $9CF$. Therefore the parameter efficiency of such method is $1/F + 1/9$ represented by the ratio of parameters after and before applying such design. Clearly, given $C$ and $F$, the parameter efficiency is always the same.

**Efficiency given the total amount of parameters.** It can be easily verified that given the total number of parameters the greatest width is reached when the best efficiency is achieved. Thus the condition for the best efficiency given the total amount of parameters should be the same with the one when the greatest width is reached.

The total number of parameters $P$ for the design can be expressed as

$$P = 3 \cdot 3 \cdot C + 1 \cdot 1 \cdot C \cdot F,$$

when studying the greatest width, we need to assume the ratio between $C$ and $F$ does not change, thus the number of output channels $F$ could be written like $F = \alpha \cdot C$ where usually $\alpha \in \mathbb{N}^+$. As a result, from the equation above when $P$ is fixed, the greatest width $G$ (i.e., $\frac{-9+\sqrt{81+4\alpha P}}{2\alpha}$) will also be fixed, which indicates that the parameter efficiency is always the same.

### 3.3.2   GROUP CONVOLUTION + POINTWISE GROUP CONVOLUTION.

**Efficiency given the input and the output.**   Similarly, we use the ratio of parameters to show parameter efficiency of this design. Given $C$ and $F$, the number of parameters after using such design can be written as $3 \cdot 3 \cdot \frac{C}{M} \cdot C + 1 \cdot 1 \cdot \frac{C}{N} \cdot F = \frac{9C^2}{M} + \frac{CF}{N}$. Since the number of parameters for standard convolution is $9CF$, the ratio will become $\frac{C}{MF} + \frac{1}{9N}$. Notice that to ensure the same size of *information field* with standard convolution, in any input group of the second layer there should be at least one output channel from each one of the output groups of the first layer, therefore $M \cdot N$ should be less than or equal to the number of output channels from the first layer, i.e., $M \cdot N \leq C$. To further illustrate the relationship between the best parameter efficiency and the choices of $M$ and $N$, we have the following theorem (the proof is given in the Appendix):

**Theorem 1.** *With the same size of* information field*, the best parameter efficiency is achieved if and only if the product of the two group numbers equals the channel number of the intermediate layer.*

As per the theorem, the best parameter efficiency can be achieved only when $M \cdot N = C$. Thus the ratio will become $\frac{N}{F} + \frac{1}{9N}$. When $F$ is a fixed number, $N$ is the only variable which could affect the efficiency. Since $\frac{N}{F} + \frac{1}{9N} \geq \frac{2}{3}\sqrt{\frac{1}{F}}$, the best efficiency can be achieved when $\frac{N}{F} = \frac{1}{9N}$, or $N = \frac{\sqrt{F}}{3}$.

**Efficiency given the total amount of parameters.**   Given the total number of parameters $P$ for one design, both $M$ and $N$ could affect the width of the network. As per Theorem 1 the greatest $C$ can be reached only when $C = M \cdot N$. When $F = \alpha \cdot C$, $P$ could be written like

$$P = 3 \cdot 3 \cdot N \cdot M \cdot N + 1 \cdot 1 \cdot M \cdot \alpha \cdot M \cdot N = MN(9N + \alpha M)$$
$$\geq MN \cdot 2\sqrt{9\alpha MN} = 6\sqrt{\alpha}C^{\frac{3}{2}}$$

Given the number of parameters $P$, width C has a upper bound when $9N = \alpha M$, which is also the condition for the best efficiency. The greatest width $G$ is $(\frac{P}{6\sqrt{\alpha}})^{\frac{2}{3}}$.

### 3.3.3   POINTWISE CONVOLUTION + DEPTHWISE CONVOLUTION + POINTWISE CONVOLUTION.

**Efficiency given the input and the output.**   Same as before, given the number of input channels $C$, bottleneck channels $K$ and output channels $F$. After applying the design, the total amount of parameters is reduced to $1 \cdot 1 \cdot C \cdot K + 3 \cdot 3 \cdot K + 1 \cdot 1 \cdot K \cdot F = K(C + F + 9)$. The number of parameters for standard convolution is still $9CF$. Notice that $K = F/4$, therefore the ratio can be further expressed as $\frac{C+F+9}{36C}$. Clearly, given $C$, $K$ and $F$, such design will also result in a fixed efficiency.

**Efficiency given the total amount of parameters.**   When $F = \alpha \cdot C$ and $K = F/4$, the total number of parameters $P$ will be

$$P = 1 \cdot 1 \cdot C \cdot \frac{\alpha C}{4} + 3 \cdot 3 \cdot \frac{\alpha C}{4} + 1 \cdot 1 \cdot \frac{\alpha C}{4} \cdot \alpha C,$$

when $P$ is fixed, the greatest width $G$ is also fixed, i.e., $\frac{-9\alpha+\sqrt{81\alpha^2+16\alpha^2 P+16\alpha P}}{2(\alpha^2+\alpha)}$.

### 3.3.4   POINTWISE GROUP CONVOLUTION + DEPTHWISE CONVOLUTION + POINTWISE GROUP CONVOLUTION

**Efficiency given the input and the output.**   We use the same way to evaluate parameter efficiency for this design. First, the number of parameters after applying such method is $1 \cdot 1 \cdot \frac{C}{M} \cdot K + 3 \cdot 3 \cdot K + 1 \cdot 1 \cdot \frac{K}{N} \cdot F = K(\frac{C}{M} + \frac{F}{N} + 9)$. The number for standard convolution is $9CF$. Since $K = F/4$

and as per Theorem 1 the best parameter efficiency can be achieved only when $K = M \cdot N$, the ratio of parameters can then be represented as $\frac{\frac{C}{M}+4M+9}{36C}$. Thus given $C$, $K$ and $F$, the best parameter efficiency can be reached by setting $\frac{C}{M} = 4M$, or $M = \frac{\sqrt{C}}{2}$.

**Efficiency given the total amount of parameters.** Similarly, according to the Theorem 1 the greatest $C$ can be reached only when the number of bottleneck channels $K = M \cdot N$. Since $F = \alpha \cdot C$ and $K = F/4$, the total number of parameters of one design $P$ can be expressed as

$$P = 1 \cdot 1 \cdot \frac{4N}{\alpha} \cdot MN + 3 \cdot 3 \cdot MN + 1 \cdot 1 \cdot M \cdot 4MN = MN(\frac{4N}{\alpha} + 9 + 4M)$$

$$\geq MN(9 + 2\sqrt{\frac{16MN}{\alpha}}) = \frac{\alpha}{4}C(9 + 4\sqrt{C})$$

Given the number of parameters $P$, the greatest width $G$ exists when $\alpha M = N$.

## 4 EXPERIMENTS

### 4.1 IMPLEMENTATION DETAILS

Table 1: Overall network layout. $B$ is the number of blocks at each stage. At the first block of each stage except the first stage down-sampling is performed and the channel number is doubled.

| Layer | Output size | KSize | Strides | Repeat |
|---|---|---|---|---|
| Image | $224 \times 224$ | | | |
| Conv1 | $112 \times 112$ | $3 \times 3$ | 2 | 1 |
| Max Pool | $56 \times 56$ | $3 \times 3$ | 2 | 1 |
| Stage 1 | $56 \times 56$ | | 1 | $B$ |
| Stage 2 | $28 \times 28$ | | 2 | 1 |
| | $28 \times 28$ | | 1 | $B-1$ |
| Stage 3 | $14 \times 14$ | | 2 | 1 |
| | $14 \times 14$ | | 1 | $B-1$ |
| Stage 4 | $7 \times 7$ | | 2 | 1 |
| | $7 \times 7$ | | 1 | $B-1$ |
| Average Pool | $1 \times 1$ | $7 \times 7$ | | 1 |
| 1000-d FC, Softmax | | | | |

The overall layout of the network is shown in Table 1. Identity mapping (He et al., 2016b) is used over each block. When building the models, we can simply replace every block in the layout with the standard convolution or the sparse kernel designs mentioned in Section 3. Batch normalization (BN) (Ioffe & Szegedy, 2015) is adopted right after each layer in the block and as suggested by (Chollet, 2016) nonlinear activation ReLU is only performed after the summation of the identity shortcut and the output of each block.

We evaluate our models on ImageNet 2012 dataset (Deng et al., 2009; Russakovsky et al., 2015), which contains 1.2 million training images and 50000 validation images from 1000 categories. We follow the same data augmentation scheme in (He et al., 2016b;a) which includes randomized cropping, color jittering and horizontal flipping. All models are trained for 100 epochs with batch size 256. SGD optimizer is used with the Nesterov momentum. The weight decay is 0.0001 and the momentum is 0.9. We adopt the similar weight initialization method from (He et al., 2015; 2016a; Huang et al., 2016). The learning rate starts with 0.1 and is divided by 10 every 30 epochs. All results reported are single center crop top-1 performances.

### 4.2 EMPIRICAL STUDY

**Relationship between the *information field* and the model accuracy.** In Section 3, we have shown that all the sparse kernel designs generated by our scheme share the same size of the *infor-*

Table 2: Comparisons to illustrate the relationship between the *information field* and the model accuracy. We tune the number of group to achieve different parameter efficiency. Width here is the number of input channels to the first stage in the network. InfoSize is the size of *information field* with regards to the input to the first stage. Numbers within the parentheses represent the number of groups. For example, GConv(1) means group convolution with only 1 group, which is also the standard convolution.

| Network Unit | #Params(×M) | Depth | Width | InfoSize | Error (%) |
|---|---|---|---|---|---|
| PW+GConv(1)+PW | 13.9 | 98 | 128 | (3, 3, 128) | 30.0 |
| PW+GConv(32)+PW | 13.9 | 98 | 256 | (3, 3, 256) | 29.2 |
| PW+GConv(1)+PW | 28.4 | 194 | 128 | (3, 3, 128) | 29.7 |
| PW+GConv(1)+PW | 28.4 | 98 | 200 | (3, 3, 200) | 29.3 |
| PW+GConv(2)+PW | 28.4 | 98 | 256 | (3, 3, 256) | 28.7 |
| PW+GConv(64)+PW | 28.4 | 98 | 512 | (3, 3, 512) | **28.4** |

*mation field* when the size of input is fixed. Meanwhile different sparse kernel designs could save different amount of parameters/computation compared to the standard convolution and the saved computation/parameters can then be used to increase the number of channels, enlarge the *information field*, and increase the final accuracy. The fundamental idea behind this is that we believe the *information field* is an essential property of all sparse kernel designs and could directly affect the final accuracy.

To verify this idea we choose a bottleneck-like design and conduct some comparisons by tuning different number of groups. We adopt the same overall network layout in Table 1. It can be easily verified that given the same size of the input tensor the change of the number of groups in the bottleneck-like design will not affect the size of the *information field* in the output. Results are shown in Table 2. Specifically, compare results on row 2 and row 5, we can see that by increasing the number of group from 2 to 32, more than a half amount of parameters will be saved to generate the same width, however the model accuracy will only decrease slightly. Meanwhile a further comparison on row 5 and row 6 indicate that if we use the saved parameters to increase the network width, the accuracy could still be improved. Since both of the two networks contain the same amount of parameters, overall network layout and type of sparse kernel design, the performance gains should only come from the increase of network width ( *information field*). Same phenomenon could also be found by comparing results on row 1 and row 2.

Besides we investigate on different usages of parameters, results on row 3 and row 4 show that the increase of network width has better potential for the improvement of accuracy than that of the depth, which also indicates that the size of the *information field* could play a more important role on the model accuracy. Additionally results in Table 2 can further explain the sparse kernel design (PW+DW+PW) in Section 3.2 where we directly apply the most parameter-efficient depthwise convolution in the middle since it has the same size of the *information field* with other group numbers.

**Comparisons of different sparse kernel designs.** We also compare different sparse kernel designs mentioned in Section 3. Results are shown in Table 3. As mentioned in Section 3 all designs have the same-sized *information field* given the same input. Results from Table 3 show that given the close amount of parameters by choosing different sparse kernel designs or group numbers models with different widths can be constructed, and the final accuracy is positively correlated to the model width (the size of the *information field*), which also coincides with our analysis above. Also notice that results here do not necessarily indicate one type of sparse kernel design is always better than the other one in terms of the parameter efficiency since as per the analysis in Section 3 the efficiency also depends on other factors like the number of groups. For example, considering the same number of parameters and overall network layout, there could be a combination of group numbers $M$ and $N$ such that the network with the design GConv($M$)+PWGConv($N$) is wider than that of DW+PW.

### 4.3 Comparisons with the State-of-the-Arts.

Based on the sparse kernel scheme, we are also able to construct more efficient designs than the state-of-the-art ones. Table 4 shows comparisons between the sparse kernel designs generated by our scheme and the state-of-the-art ones. For fair comparisons, we use the same network layout as shown in Table 1 and replace blocks in it with corresponding designs, and the model size around 11.0M is selected as it is the size that different models (e.g., Xception, ResNeXt and ShuffleNet) can be

Table 3: Comparisons of different sparse kernel designs. All designs share the same network layout.

| Network Unit | #Params($\times$M) | Width | InfoSize | Error (%) |
|---|---|---|---|---|
| Standard Convolution | 11.2 | 64 | (3, 3, 64) | 31.1 |
| DW+PW | 0.8 | 72 | (3, 3, 72) | 31.7 |
| DW+PW | 11.2 | 280 | (3, 3, 280) | 28.5 |
| GConv(4)+PWGConv(32) | 11.2 | 128 | (3, 3, 128) | 30.8 |
| GConv(16)+PWGConv(16) | 11.3 | 256 | (3, 3, 256) | 29.4 |
| PW+DW+PW | 11.0 | 400 | (3, 3, 400) | 26.9 |
| PWGConv(4)+DW+PWGConv(4) | 11.3 | 560 | (3, 3, 560) | **25.6** |

Table 4: Comparisons with different state-of-the-art sparse kernel designs. All settings are restored from the original papers. Specifically, bottleneck ratio is $1 : 4$ for ResNet and ResNeXt adopts cardinality of 16 and bottleneck ratio of $1 : 2$. Meanwhile 4 groups are used for ShuffleNet.

| Network Unit | #Params($\times$M) | Width | InfoSize | Error (%) |
|---|---|---|---|---|
| ResNet (He et al., 2016a) | 11.2 | 64 | (3, 3, 64) | 31.3 |
| ResNet with bottleneck (He et al., 2016a) | 11.3 | 192 | (3, 3, 192) | 29.9 |
| ResNeXt (Xie et al., 2017) | 11.1 | 192 | (3, 3, 192) | 29.8 |
| Xception (Chollet, 2016) | 11.2 | 280 | (3, 3, 280) | 28.5 |
| ShuffleNet (Zhang et al., 2017) | 11.3 | 560 | (3, 3, 560) | 25.6 |
| GConv(100)+PWGConv(2) | 8.6 | 200 | (3, 3, 200) | 27.0 |
| PWGConv(100)+DW+PWGConv(2) | 10.4 | 700 | (3, 3, 700) | **24.9** |

easily configured to. Results in Table 4 indicate that sparse kernel designs in our scheme could even yield better accuracy with a smaller model size, which also validates the idea of our sparse kernel scheme. Also notice that the choices of group numbers used in our designs are chosen to help easily accommodate both the similar model size and the overall network layout, which may not be the most efficient ones that are supposed to result in a wider network with better accuracy under the same limitation of parameters.

## 5 RELATED WORKS

**Model Compression.** Traditional model compression techniques include pruning, quantization and low-rank approximation. Pruning (Wen et al., 2016; Ardakani et al., 2016; Liu et al., 2017; Li et al., 2016; He et al., 2017; Liu et al., 2015) reduces redundant weights, network connections or channels in a pre-trained model. However, it could face difficulty for deploying on hardware like GPU since some pruning methods may be only effective when the weight matrix is sufficiently sparse. Quantization (Zhou et al., 2016; 2017; Courbariaux et al., 2015; 2016; Deng et al., 2018; Micikevicius et al., 2017) reduces the number of bits required to represent weights. Unfortunately, this technique will require specialized hardware support. Low rank approximation (Lebedev et al., 2014; Jin et al., 2014; Wang et al., 2016; Xue et al., 2014; Novikov et al., 2015; Garipov et al., 2016) uses two or more matrices to approximate the original matrix values in a pre-trained model. Nevertheless, since the process is an approximation of original matrix values maintaining a similar accuracy will always need additional re-training. The focus of this paper, the sparse kernel approach, mitigates all these problems by directly training networks using structural sparse convolutional kernels.

## 6 CONCLUSION

In this paper, we present a scheme to craft the effective sparse kernel design by eliminating the large design space from three aspects: composition, performance and efficiency. During the process to reduce the design space, we find an unified property named *information field* behind various designs, which could directly indicate the final accuracy. Meanwhile we show the final 4 designs in our scheme along with detailed efficiency analysis. Experimental results also validate the idea of our scheme.

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

## 7 APPENDIX

**Proof of Theorem 1**

*Proof.* Without loss of generality we use the example in Section 3.3.2 to prove the theorem. Recall that the total number of parameters for such design can be expressed as

$$P = 3 \cdot 3 \cdot \frac{C}{M} \cdot C + 1 \cdot 1 \cdot \frac{C}{N} \cdot F = \frac{9C^2}{M} + \frac{CF}{N},$$

then the problem could be interpreted as proving that the minimum value of $P$ can be achieved if and only if $M \cdot N = C$.

We prove the theorem by contradiction. Assume the minimum value of $P$ could be achieved when $M \cdot N < C$. Then we can always find a $N' = C/M > N$ such that the combination of $M$ and $N'$ could result in a smaller value of $P$, which contradicts our assumption. The theorem is hence proved. □

