# OpenReview forum: "Penetrating the Fog: the Path to Efficient CNN Models"
_ICLR.cc/2019/Conference_

### Official Review · AnonReviewer3 · 2018-11-02
**this paper gives a recipe for efficient CNN, especially tailored for mobile platforms**

**Rating:** 5
**Confidence:** 3

**Review:**

The paper considers sparse kernel design in order to reduce the space complexity of  a convolutional neural network. In specifics, the proposed procedure is composed of following steps: 1) remove repeated layers, 2) remove designs with large degradation design, and 3) further remove design for better parameter efficiency.

The paper proposed the composition of group convolution, pointwise convolution, and depthwise convolution  for the sparse kernel design, which seems pretty promising. In addition, the authors discussed the efficiency of each convolution compositions.

I failed to appreciate the idea of information field, I didn't understand the claims that "For one output tensor, sizes of information fields for all activations are usually the same". When introducing a new concept, it's very important to make it clear and friendly. The author could consider more intuitive, high level, explanation, or some graphic demonstrations. Also, I couldn't see why this notion is important in the rest of the paper.

Personally I'm so confused by the theorem. It looks like a mathematical over-claim to me. It claims that the best efficiency is achieved when M N = C. However, is it always the case? What is M N \neq C? What does the theorem mean for real applications?

All the reasoning and derivation are assuming the 3 x 3 spatial area and 4 way tensor. I would assume these constant are not important, the paper could be much stronger if there is a clear notion of general results.

---

### Official Review · AnonReviewer1 · 2018-11-05
**A good start**

**Rating:** 4
**Confidence:** 3

**Review:**

Standard dense 2D convolution (dense in space and channels) may waste parameters. This paper points out the many ways that sparser convolutional operators (“kernels”) may be combined into a single combined operator that may be used in place of dense convolution.

The paper waxes grandiose about the exponentially many ways that operations may be combined but then defines and tries only four. While trying four alternatives may be quite interesting, the paper could have avoided grandiose language by just stating: “We tried four things. If you restrict yourself to kernels with 3x3 receptive field and no repeated operations <and probably other assumptions>, there are only four unique combinations to be tried.” Perhaps a page of text could have been saved.

The paper also defines “information field” as the product of the operator’s (spatial) receptive field and the number of channels that each unit can see. Authors proceed to make broad claims about how information field is an important concept that predicts performance. While this may indeed turn out to be an important concept, it is not shown as such by the paper.

Claims:

“…we identify a easily measurable quantity named information field behind various sparse kernel designs, which is closely related to the model accuracy.”

“During the process to reduce the design space, we find an unified property named information field behind various designs, which could directly indicate the final accuracy.”

But the paper does not substantiate these claims.

Since information field is defined as the product of the receptive field and the number of channels seen, it would seem necessary to show, say, at least some experiments with varying receptive field sizes and number of channels. Then it might be shown, for example, that across a wide array of network sizes, widths, depths, holding all but information field constant, information field is predictive of performance. But these experiments are not done.

Receptive fields: the paper *only ever tries 3x3 receptive fields* (Table 2, 3, 4). So absolutely no support is given for the relevance of two out of the three components (i size, j size) comprising information field!

Number of channels: as far as I can tell, Table 2 and 3 contain the only results in this direction. Reading off of Table 2: for networks of the same depth (98), info size 256 works a bit better than 128*, and 512 works a bit better than 256.

* (see also Table 3 lines 4 vs 5 show the same 256 vs 128 effect.)

Cool. But *two comparisons* are not even close to enough to support the statement “we find an unified property named information field behind various designs”. It is enough to support the statement “for this single network we tried and using 3x3 receptive fields, we found that letting units see more channels seemed to help.” Unfortunately, this conclusion on its own is not a publishable result.



To make this paper great, you will have to close the gap between what you believe and what you have shown.

(1) You believe that information field is predictive of accuracy. So show it is predictive of accuracy across sufficiently many well-controlled experiments.

(2) It may also be that the PWGConv+DW+PWGConv combination is a winning one; in this case, show that swapping it in for standard convolution helps in a variety of networks (not just ResNet) and tasks (not just ImageNet).



Other minor notes:

 - Equations are critical in some parts of some papers, but e.g. triple nested sums probably aren’t the easiest way of describing group convolution.

 - The part about regexes seemed unnecessary. If 1000 different designs were tried in a large automated study where architectures were generated and pruned automatically, this detail might be important (but put it in SI). But if only four are tried this detail isn’t needed: we can see all four are different.

 - Figure 1 is a great diagram!

 - How efficient are these kernels to compute on the GPU? Include computation time.

 - “Efficiency given the total amount of parameters.” These equations and scaling properties seemed to miss the point. For example, “It can be easily verified that given the total number of parameters the greatest width is reached when the best efficiency is achieved.” This is just saying that standard convolution scales poorly as F -> infinity. This doesn’t seem like the most useful definition of efficiency. A better one might be “How many params do you need to get to x% accuracy on ImageNet?” Then show curves (# of params vs accuracy) for variants of a few popular model architectures (like ResNet or Xception with varying width and depth).

 - 3.3.2: define M and N

---

### Official Review · AnonReviewer2 · 2018-11-08
**Interesting topic but the paper is not well explained**

**Rating:** 5
**Confidence:** 3

**Review:**

This paper addressed an interesting problem of reducing the kernel to achieve CNN models, which is important and attracts lots of research work. However, the methods don't have very good justifications.
For example, in Section 3.1, the authors mentioned that "Specifically, in normal CNNs it is quite common to have multiple stages/blocks which contain repeated patterns such as layers or structures." It is still unclear why it is better to replace these so-called repeated patterns.
The defined "information field" is not clearly explained and the benefits are also not demonstrated.

---

### Meta-Review · Area_Chair1 · 2018-12-17
**lack of support**

**Confidence:** 5
**Recommendation:** Reject

**Metareview:**

This paper points out methods to obtain sparse convolutional operators. The reviewers have a consensus on rejection due to clarity and lack of support to the claims.